# Effect of Serum Albumin on Porphyrin-Quantum Dot Complex Formation, Characteristics and Spectroscopic Analysis

**DOI:** 10.3390/nano11071674

**Published:** 2021-06-25

**Authors:** André L. S. Pavanelli, Leandro N. C. Máximo, Roberto S. da Silva, Iouri E. Borissevitch

**Affiliations:** 1Departamento de Física, Faculdade de Filosofia, Ciências e Letras de Ribeirão Preto, Universidade de São Paulo, Av. Bandeirantes 3900, Ribeirão Preto 14040-900, Brazil; andpavanelli@gmail.com; 2Instituto Federal de Educação, Ciência e Tecnologia Goiano, Urutaí 75790-000, Brazil; lncmaximo@gmail.com; 3Departamento de Ciências Biomoleculares, Faculdade de Ciências Farmacêuticas de Ribeirão Preto, Universidade de São Paulo, Av. Bandeirantes 3900, Ribeirão Preto 14040-900, Brazil; silva@usp.br

**Keywords:** CdTe-3-MPA quantum dot, TPPS_4_ and TMPyP porphyrins, quantum dot-porphyrin complex, bovine serum albumin effects, pH effect

## Abstract

The effect of bovine serum albumin (BSA) upon interaction between CdTe QD functionalized by 3-Mercaptopropionic Acid (CdTe-3-MPA QD) and two water soluble porphyrins: positively charged *meso*-tetra methyl pyridyl porphyrin (TMPyP) and negatively charged *meso*-tetrakis(p-sulfonato-phenyl) porphyrin (TPPS_4_), was studied in function of pH using the steady-state and time resolved optical absorption and fluorescence spectroscopies. It was shown that, depending on the charge state of the components, interaction with albumin could either prevent the formation of the QD…PPh complex, form a mixed QD…PPh…BSA complex or not affect PPh complexation with QD at all. The obtained results may be of interest for application in photomedicine.

## 1. Introduction

Photodynamic therapy (PDT) is a method of treatment of various diseases, including several types of cancer [1,2,3,4,5,6]. PDT is based on mutual application of a photoactive compound (photosensitizer, PS), visible or infrared light and molecular oxygen to generate reactive oxygen species (ROS). The efficacy of PDT depends on the PS structural characteristics, which are important to increase its uptake and subcellular localization. Porphyrin-type compounds are of a special interest among PS. Due to their intensive optical absorption in the required spectral regions, high quantum yield of the triplet state and relative intense fluorescence, high dark and photostability and affinity with biological structures, such as proteins, nucleic acids and cell membranes, porphyrins (PPh) are widely applied as PS in Photodynamic Therapy (PDT) [1,2,3,4,5,6] and as fluorescence probes in Fluorescence Diagnostics (FD) [7,8]. However, a great disadvantage of PPh for these applications lies in their relatively low optical absorption in the spectral region of the so-called phototherapeutic window (600–800 nm), where biological tissues are the most transparent. The problem could be corrected by PPh complexation with certain structures, able to absorb light energy in this spectral range and transfer this energy to PPh molecules.

Semiconductor nanocrystals (quantum dots, QD) possess intense optical absorption in a wide spectral range and intensive and narrow fluorescence band, the position of which depends on QD core size [9]. Due to these extraordinary characteristics, QD can serve as an antenna, accumulating the light energy in a wide spectral region and passing it to other compounds, PPh, in particular [10,11,12,13].

On the other hand, being inorganic semiconductors, QDs manifest high toxicity toward biological objects and possess low water solubility, which restricts their direct application in medicine and biology. However, when the QD surface is covered with appropriate molecules (functionalized QD), it overcomes this imperfection, as in this case living systems appear protected from the contact with the toxic QD core (CdTe or CdSe, for example). Moreover, functionalization of the QD surface by specific groups, for example, aptamers, can increase their affinity to specific biological targets, malignant tissues in particular [14,15]. This makes QD a promising drug delivery system [16,17,18].

At the same time, the mechanisms of QD action in photochemotherapy, not well studied yet, differ from the photodynamic effect, the basis of the photodynamic therapy (PDT), which is effective and well described. Therefore, the complexes of porphyrins with quantum dots are of great interest for medicine. PPh and QD complexation can occur via energy, electron or proton transfer, depending on the structural characteristics of the participants [19,20,21,22,23,24,25,26,27,28,29].

When introduced into an organism, PPh and QD should interact with various organized structures, such as cell membranes, nucleic acids, proteins, etc. Among them, serum albumin attracts a special attention due to its ability to bind various compounds and transport them within the organism with blood flux [30]. Besides, albumins possess elevated affinity to malignant tissues [31,32]. This makes albumins perspective compounds as delivery systems. On the other hand, interaction with albumins can affect complexation of PPh with QD, changing its and the complex characteristics.

In the present work, we have studied the effect of bovine serum albumin on interaction between CdTe QD functionalized by 3-mercaptopropionic acid (CdTe-3-MPA QD) and two water soluble porphyrins: positively charged *meso*-tetra methyl pyridyl porphyrin (TMPyP) and negatively charged *meso*-tetrakis(p-sulfonato-phenyl) porphyrin (TPPS_4_). The study was realized at pH 7.0 and pH 4.0 using optical absorption and fluorescence spectroscopies. It was shown that, depending on the charge state of components, interaction with albumin could produce various effects: prevent formation of the QD…PPh complex, form a mixed QD…PPh…BSA complex or not affect the PPh complexation with QD at all. The obtained results may be of interest for application in photomedicine.

## 2. Materials and Methods

TMPyP and TPPS_4_ porphyrins (Figure 1) were purchased from Mid Century Chemicals (Posen, IL, USA), and used without any additional purification. Bovine serum albumin (BSA) was obtained from Sigma Aldrich Products (San Luis, MO, USA). CdTe-3-MPA QD were synthesized at the Department of Biomolecular Sciences of the Faculty of Pharmaceutic Sciences, Sao Paulo University, Brazil. The synthesis was realized in an aqueous media from Cd^2+^ and Te^2−^ solutions in accordance with the procedure described in details in [33] with some modifications. The solution was prepared by adding the oxygen-free NaHTe solution to the Cd^2+^ precursor solution at pH 11, in the absence of light, under continuous argon flow.

The QD diameter (*D*) and molar absorption coefficient (*ε*), were calculated by the Equations (1) and (2) [34]:(1)$$D=\left(9.8127\times 10^{-7}\right)\lambda^{3}-\left(1.7147\times 10^{-3}\right)\lambda^{2}+\left(1.0064\right)\lambda -194.84$$
(2)$$\varepsilon =10043\;{\left(D\right)}^{2.12}$$
leading to *D* ≈ 2.5 nm and *ε*_λ557 nm_ = 7.0 × 10^5^ M^−1^ cm^−1^, respectively.

The compound concentrations were determined from optical absorption (*A*) as
(3)$$C=\frac{A}{\varepsilon }$$
using molar absorption coefficients: for BSA *ε* = 4.55 × 10^4^ M^−1^ cm^−1^ at *λ* = 280 nm, for CdTe-3-MPA QD *ε* = 1.27 × 10^5^ M^−1^ cm^−1^ at *λ* = 557 nm, for TMPyP *ε* = 2.26 × 10^5^ M^−1^ cm^−1^ at *λ* = 425 nm, for non-protonated TPPS_4_ form (pH 7.0) *ε* =1.3 × 10^4^ M^−1^ cm^−1^ at *λ* = 515 nm and *ε* = 3.26 × 10^4^ M^−1^ cm^−1^ at *λ* = 644 nm for its biprotonated form (pH 4.0).

### 2.1. Steady-State Optical Absorption and Fluorescence Measurements

The absorption spectra were obtained with the help of a Beckman Coulter DU-640 spectrophotometer and the fluorescence spectra were monitored by a Hitachi 7000 spectrometer.

To exclude the effect of changes in optical absorption at excitation wavelengths, the fluorescence intensities obtained experimentally were corrected in accordance with the equation:(4)$$I_{corr}=\frac{I_{exp}}{A_{ex}}$$
where $I_{exp}$ and $I_{corr}$ are experimental and corrected fluorescence intensities, respectively, and $A_{ex}$ is the absorbance at the excitation wavelength.

### 2.2. Measurements of Time Resolved Fluorescence

Time-resolved fluorescence experiments were made using the time-correlated single photon counting technique. The excitation source was a pulse titanium-sapphire laser Tsunami 3950 pumped by a Millena Xs laser, both from Spectra Physics, with 5 ps pulse width at the half height and 8.0 MHz frequency, controlled by the pulse picker 3980 from Spectra Physics. The excitation wavelengths were obtained by the BBO (GWN-23PL from Spectra Physics) crystal. The measurements were made with a FL9000 spectrometer from Edinburgh, adjusted in an ‘L’ configuration with the excitation source. The wavelengths for measurement were selected by a monochromator and the detection was made by a Hamamatsu R3809U photomultiplier. The average time response of the instrument was 100 ps.

The PPh, QD and BSA stock solutions were prepared in phosphate buffer (pH 6.8, 7.5 mM). All experiments were realized at room temperature (24 ± 1) °C.

The experimental data were treated using OriginPro 8 commercial program. All final values were average of three independent experiments.

## 3. Results and Discussion

The CdTe-3-MPA QD possess optical absorption in the spectral range from 200 nm to 650 nm with an accentuated peak in the range from 425 nm to 575 nm with *λ*_max_ = 490 nm (Figure 2A). The fluorescence emission spectra of the synthesized CdTe-3-MPA QD were measured with excitation at 490 nm. Figure 2B shows the emission spectra with a band in the region of 500 nm to 700 nm (*λ*_max_ = 585 nm).

### 3.1. Quantum Dot Interaction with BSA

The results presented in this section were obtained at pH 7.0. At pH 4.0 we have observed no significant effect of BSA on QD characteristics in the concentration range used.

In an aqueous solution, BSA possesses UV-visible spectrum with maximum absorption at 280 nm and the fluorescence emission spectrum with maximum at 380 nm when excited at 280 nm. Interaction of BSA with CdTe-3-MPA QD in aqueous solutions show significant increase of the QD stability with no precipitation even after 24 h. The QD luminescence decay curves, registered for 480 nm excitation have been fitted successfully as the sum of two exponents (Figure 2C):(5)$$I=I_{1}\times \exp(-t/\tau_{1\mathrm{QD}\_\mathrm{BSA}})+I_{2}\times \exp(-t/\tau_{2\mathrm{QD}\_\mathrm{BSA}})$$

With lifetimes of the components $\tau_{1\mathrm{QD}\_\mathrm{BSA}}$ = (0.9 ± 0.1) ns and $\tau_{2\mathrm{QD}\_\mathrm{BSA}}$ = (20 ± 1) ns.

Addition of BSA does not affect the QD absorption spectrum, however, it increases the QD luminescence intensity (Figure 2B, inset). The profile of the QD luminescence decay curve continues biexponentially with invariable lifetimes of the components, and their relative amplitudes vary with the BSA concentration (Table 1).

The contribution of the respective component can be calculated as the ratio between its amplitude (*I*_1_ or *I*_2_) and the sum of amplitudes of both components (*I*_1_ + *I*_2_). Therefore, the contribution of the long-lived component, calculated as
(6)$$R_{2}=\frac{I_{2}}{I_{1}+I_{2}}$$
where *I*_1_ and *I*_2_ are the amplitudes of the short-lived and the long-lived components, respectively, and suffer an increase, which coincides with the increase in the integral luminescence intensity (compare the insets on Figure 2B,C).

Thus, it is possible to conclude that the increase of the QD luminescence intensity may be associated with increase of the contribution of the long-lived component $R_{2}$. Consequently, the contribution of the short-lived component decreases.

As it has been shown recently [13,26,27,28,29], the short-lived component of the CdTe-3-MPA QD luminescence is associated with the electron-hole annihilation in the QD core, while the long-lived one is due to the electron-hole annihilation in the QD functionalizing (protective) shell. Therefore, the increase of the QD luminescence intensity in the presence of BSA can be due to interaction of BSA with 3-MPA groups on the QD surface.

The binding constant of QD with BSA (*K*_QD_BSA_) was calculated using the equation from [35]:(7)$$\frac{1}{I-I_{min}}=\frac{1}{I_{max}-I_{min}}+\frac{1}{(I_{max}-I_{min})K_{\mathrm{QD}\_\mathrm{BSA}}\;}\frac{1}{\left[BSA\right]}$$
where *I*_min_, *I* and *I*_max_ are integral luminescence intensities in the absence of BSA, for BSA concentration [BSA] and for the BSA excess.

The determined value was *K*_QD_BSA_ = (1.2 ± 0.1) × 10^6^ M^−1^.

The fact that at pH 4.0 no changes in the QD and BSA characteristics at used concentrations were observed may be explained by strong electrostatic repulsion between positively charged QD and BSA. Indeed, the isoelectric point (IP) of BSA is at pH 4.7 [30]. At the same time the pK_a_ point of CdTe-3-MPA QD is at pH 4.34 [36]. Thus, at pH 4.0 the total charge of both BSA and QD is positive, which should reduce the probability of their binding making the binding constant much less than (1.2 ± 0.1) × 10^6^ M^−1^ observed for pH 7.0.

### 3.2. Effect of BSA upon QD Interaction with TMPyP Porphyrin

Due to the presence of four positive side groups in its structure TMPyP porphyrin possesses 4+ charge in the pH range from 2.0 up to 9.0. These series of experiments were made at pH 4.0 and pH 7.0. However, the results, obtained for these two pHs were quite similar. Therefore, we present here only the results obtained in aqueous phosphate buffer solution at pH 7.0.

Interaction between CdTe-3MPA QD and TMPyP porphyrin in water solutions is characterized by formation of a charge transfer complex between negatively charged MPA groups of QD and positively charged methyl-pyridinium groups of TMPyP [28]. The complex is characterized by optical absorption peaks at 455 nm, 585 nm and 629 nm and a fluorescence peak at 632 nm (Figure 3). The fluorescence decay curve is monoexponential with the lifetime *τ*_TMPyP_QD_= (5.3 ± 0.2) ns (Figure 3, inset).

The binding constant, determined from TMPyP fluorescence quenching by QD was *K*_TMPyP_QD_ = (6.0 ± 0.5) × 10^6^ M^−1^ [28].

The addition of BSA to the solution with formed CdTe-3MPA…TMPyP complex induces changes neither in the complex absorption and fluorescence spectra nor in the profile of the fluorescence decay curve (Figure 3).

Another series of experiments was made adding QD into the solution containing TMPyP and BSA. It was shown formerly [37] that TMPyP and BSA form a complex characterized by the optical absorption spectrum with Soret peak at 425 nm and three peaks in the Q region (Figure 4A) and fluorescence spectrum with a peak at 629 nm (Figure 4B). The binding constant of TMPyP with BSA was estimated as *K*_TMPyP_BSA_ = 7.3 × 10^5^ M^−1^ [37].

The addition of QD into the TMPyP + BSA solution induces the reduction of the absorption peak at 425 nm and formation of a new absorption peak at 455 nm. This new absorption peak coincides with that of CdTe-3MPA…TMPyP complex (see Figure 3). Besides, the QD addition induces increase of the solution fluorescence intensity. Thus, we can associate these spectral changes with the formation of CdTe-3MPA…TMPyP charge transfer complex even in the BSA presence.

Binding constant of TMPyP with QD in the presence of BSA was calculated from the dependence of the integral fluorescence on the QD concentration using the Equation (7), which led to *K*_TMPyP_QD + BSA_ = (10 ± 3) × 10^6^ M^−1^, close to that obtained in the BSA absence.

Thus, we can affirm that BSA in the solution does not destruct CdTe-3MPA…TMPyP complex already formed, and does not reduce the probability of its formation, as well. This could be explained by the fact that the constant of the CdTe-3MPA…TMPyP complex formation is approximately 10 times larger than that for TMPyP binding with BSA and 5 time larger than for CdTe-3MPA binding with BSA. Therefore, in the system of equilibria
CdTe-3MPA + TMPyP ⇔ CdTe-3MPA…TMPyP   *K*_TMPyP_QD_ = (6.0 ± 0.5) × 10^6^ M^−1^
BSA + TMPyP ⇔ BSA…TMPyP   *K*_TMPyP_BSA_ = 7.3 × 10^5^ M^−1^
CdTe-3MPA + BSA ⇔ CdTe-3MPA…BSA      *K*_QD_BSA_ = (1.2 ± 0.1) × 10^6^ M^−1^
the first one should be predominant.

Taking into account that IP of BSA is at pH 4.7, at pH 7.0 its net charge is negative and the electrostatic attraction should stimulate its binding with positively charged TMPyP molecules. However, TMPyP interaction with CdTe-3MPA QD appears stronger. This may be due to stabilization of TMPyP-QD contact via charge transfer complex formation.

The question is whether the CdTe-3MPA…TMPyP complex can bind with BSA. However, no spectral changes have been observed which could be associated with the formation of this complex.

### 3.3. Effect of BSA on Interaction of QD with TPPS_4_ Porphyrin

Due to the presence of four nitrogen atoms in the porphyrin ring structure TPPS_4_ porphyrin can be biprotonated, with the pK_a_ point close to pH 5.0 [38]. Therefore, at pH 7.0, where TPPS_4_ is in the non-protonated form, it possesses a charge 4^−^, while at pH 4.0, where it is in the biprotonated form, its charge is 2^−^. This determines the difference in the TPPS_4_ behavior at these two pHs. Therefore, at pH 7.0 the non-protonated TPPS_4_ is characterized by the optical absorption spectrum with Soret peak at 413 nm and four peaks in the Q-region, the largest localized at 518 nm, and fluorescence with maximum at 640 nm (Figure 5) and the lifetime *τ*_pH7_ = 8.5 ns. Biprotonated TPPS_4_ possesses the absorption spectrum with Soret peak at 434 nm and three peaks in the Q-region with the largest one at 646 nm and fluorescence with maximum at 665 nm (Figure 5) and the fluorescence lifetime of *τ*_pH4_ = 3.6 ns [39].

#### 3.3.1. Effect of BSA on Interaction of QD with TPPS_4_ Porphyrin at pH 7.0

Recently we have demonstrated [29] that at pH 7.0 interaction of TPPS_4_ with CdTe-3-MPA QD is realized via energy transfer from QD to TPPS_4_. This interaction induces no changes in absorption and emission spectra of both TPPS_4_ and QD; however, it reduces the QD luminescence intensity. The binding constant of the non-protonated TPPS_4_ with QD was estimated as *K*_TPPS_QD pH7_ = (2.5 ± 0.1) × 10^6^ M^−1^ [29].

Interaction of TPPS_4_ with BSA has recently been studied in [37]. Addition of BSA into the TPPS_4_ solutions at pH 7.0 induces the shift of the TPPS_4_ Soret absorption band from *λ* = 412 nm to *λ* = 422 nm and the fluorescence peak from *λ* = 645 nm to *λ* = 654 nm. The TPPS_4_ fluorescence intensity increases (Figure 6), and lifetime is practically unchanged *τ*_TPPS_BSA pH7_ = (10.8 ± 0.1) ns. The TPPS_4_-BSA binding constant is *K*_TPPS_BSA pH7_ = 3.2 × 10^6^ M^−1^ [37].

The addition of BSA into the TPPS_4_…QD solution shifts the TPPS_4_ absorption band from 412 nm to 422 nm and a new band centered at 436 nm appears. Simultaneously, the TPPS_4_…QD complex emission increases and an emission peak at 650 nm is formed (Figure 7).

These new peaks combine with no peak of either pure TPPS_4_, pure CdTe-3-MPA QD or TPPS_4_…CdTe-3-MPA complex. Therefore, we believe they reflect formation of a new mixed TPPS_4_…CdTe-3-MPA…BSA complex. The binding constant calculated from the fluorescence data in accordance with the Equation (7) is *K*_TPPS_QD + BSA pH7_ = (6.5 ± 0.2) × 10^6^ M^−1^.

Addition of QD into the TPPS_4_ + BSA mixed solution induces no changes in the absorption and fluorescence spectra of these solutions, demonstrating that binding of TPPS_4_ with BSA at pH 7.0 prevents the formation of TPPS_4_…CdTe-3-MPA complex and consequently the formation of the mixed TPPS_4_…CdTe-3-MPA…BSA complex.

Effective binding of negatively charged non-protonated TPPS_4_ with BSA, which at pH 7.0 is also negatively charged, shows that in this case electrostatic repulsion is not dominant and other types of interaction, such as a hydrophobic one, should be responsible for porphyrin binding with BSA. High stability of the TPPS_4_…CdTe-3-MPA complex cannot be associated with the electrostatic interaction, as well, since at pH 7.0 the MPA groups of QD are non-protonated and do not possess a positive charge. The effective contribution of non-electrostatic interactions in the non-protonated TPPS_4_ + QD + BSA system follows from the fact of formation of the mixed TPPS_4_…QD…BSA complex. However, more detailed and justified conclusions about the nature and contributions of various types of interaction in this complex system require more profound studies using other experimental techniques besides spectroscopic ones.

#### 3.3.2. Effect of BSA on Interaction of QD with TPPS_4_ Porphyrin at pH 4.0

As it has been demonstrated earlier [29], at pH 4.0 interaction of TPPS_4_ with CdTe-3-MPA QD occurs via two mechanisms: the proton transfer from TPPS_4_ to QD and the energy transfer from QD to TPPS_4_, the first one being predominant. The proton transfer changes the TPPS_4_ absorption and fluorescence spectra, characteristic for the biprotonated TPPS_4_, to those of the non-protonated form (Figure 8). The fluorescence lifetime changes from $\tau_{\mathrm{TPPS}\;\mathrm{pH}4}=\left(3.6\pm 0.2\right)\;\mathrm{ns}$ to $\tau_{\mathrm{TPPS}\_\mathrm{BSA}\;\mathrm{pH}4}=\left(8.5\pm 1\right)\;\mathrm{ns}$, which is also characteristic for the non-protonated TPPS_4_.

The binding constant of TPPS_4_ with CdTe-3-MPA QD, calculated from fluorescence data is $K_{\mathrm{TPPS}\_\mathrm{QDpH}4}=\left(2.4\pm 0.2\right)\times 10^{6}$ M^−1^ [29].

Addition of BSA into the TPPS_4_ solutions at pH 4.0 induces the TPPS_4_ deprotonation, as well, leading to spectral characteristics equal to those of the non-protonated TPPS_4_ bound with BSA: Soret peak at λ = 422 nm, fluorescence peak at 654 nm and fluorescence lifetime *τ* = 10.8 ns. The obtained binding constant of TPPS_4_ with BSA is *K*_TPPS_BSA pH4_ = 1.5 × 10^8^ M^−1^ [37].

Addition of BSA into the TPPS_4_…CdTe-3-MPA complex solution changes the profile and peak positions of the optical absorption and fluorescence spectra in such a way that the final ones coincide with those of TPPS_4_ bound with BSA (Figure 9). The fluorescence intensity and lifetime increase (Figure 9, inset), which is also characteristic for TPPS_4_ binding with BSA. This could mean that addition of BSA destructs the TPPS_4_…CdTe-3-MPA complex in favor of the TPPS_4_ complex with BSA. At the same time, the calculation of the TPPS_4_ binding constant with BSA using the Equation (3) leads to $K_{{\mathrm{TPPS}}_{\mathrm{QD}}+\mathrm{BSApH}4}=\left(1.2\pm 0.2\right)\times 10^{10\;}$ M^−1^ (Figure 10). This value is ≈100 times larger than that for TPPS_4_ binding with BSA in the absence of CdTe-3-MPA QD. This demonstrates that the TPPS_4_ binding with QD increases the probability of its binding with BSA.

When QD is added into the TPPS_4_ + BSA solutions, no changes either in its optical absorption and fluorescence spectra or in the fluorescence lifetime occur, which demonstrates that the porphyrin binding with BSA blocks its binding with QD.

## 4. Conclusions

Depending on the porphyrin characteristics, principally its charge state, its interaction with bovine serum albumin (BSA) manifests in different forms.

In the case of *meso*-tetra methyl pyridyl porphyrin (TMPyP), which possesses charge 4+ and forms a charge transfer complex with CdTe-3-MPA quantum dots, albumin neither destructs this complex already formed nor reduces the probability of its formation.

For *meso*-tetrakis(p-sulfonato-phenyl) porphyrin (TPPS_4_), non-protonated at pH 7.0, which has a charge 4− and forms an energy transfer complex with CdTe-3-MPA quantum dots, BSA is able to form a mixed TPPS_4_…CdTe-3-MPA…BSA complex.

In the case of the biprotonated TPPS_4_ with charge 2− at pH 4.0, for which the complexation with CdTe-3-MPA includes both an energy transfer and a proton transfer, BSA destructs this complex in favor of the TPPS_4_…BSA one and prevents TPPS_4_ from complexation with QD. Moreover, surprisingly, TPPS_4_ complexation with QD increases the probability of its binding with BSA.

We consider this study as the first step in investigation of effects of albumins upon the porphyrin complexation with quantum dots. To clarify the mechanisms of these effects, more detailed studies are necessary with the help of various experimental techniques, besides the spectroscopic ones. However, since the main purpose of this study is to evaluate the perspective of application of porphyrin…quantum dot complexes in photochemotherapy, which is based on light exposure, the spectroscopic results presented in this paper can be important for application of porphyrin…quantum dot complexes in medicine for fluorescence diagnostics and photodynamic therapy.

## Figures and Tables

**Figure 1 nanomaterials-11-01674-f001:**
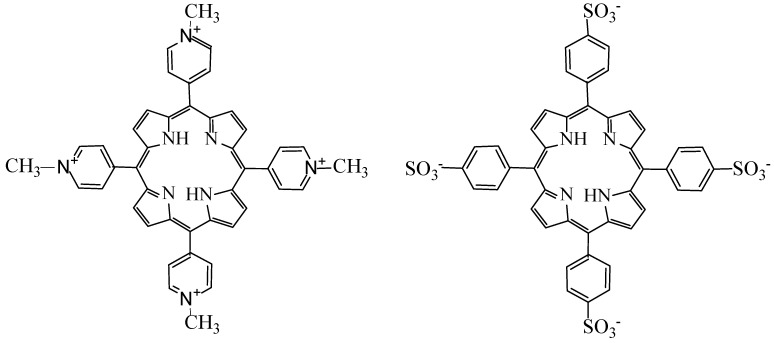
Chemical structures of TMPyP (**left**) and TPPS_4_ (**right**).

**Figure 2 nanomaterials-11-01674-f002:**
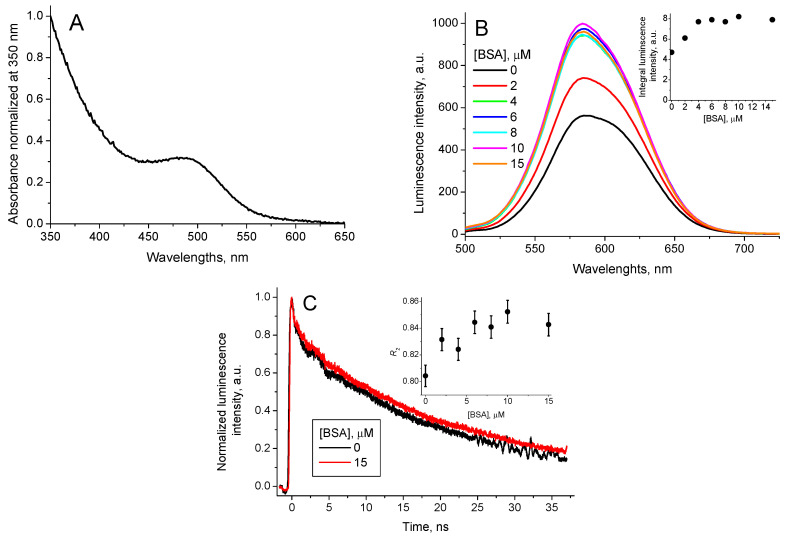
CdTe-3-MPA QD (**A**) absorption spectrum, normalized by absorbance at 350 nm, (**B**) QD luminescence spectrum (*λ*_ex_ = 490 nm) for different BSA concentrations, (inset: the integral QD luminescence intensity in function of the BSA concentration) and (**C**) QD luminescence decay curves (*λ*_ex_ = 490 nm, *λ*_em_ = 580 nm) for [BSA] = 0 and [BSA] = 15 μM (inset: the contribution of the long-lived luminescence decay component *R_2_* in function of the BSA concentration).

**Figure 3 nanomaterials-11-01674-f003:**
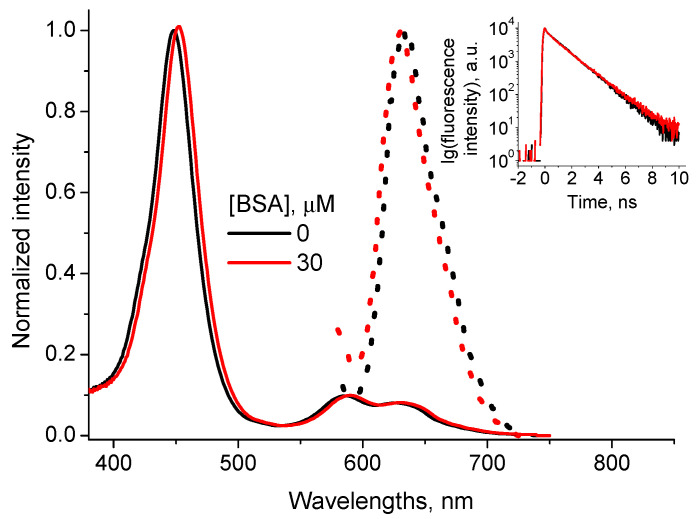
Normalized optical absorption (solid lines) and fluorescence (points) spectra (λ_ex_ = 455 nm) of the CdTe-3MPA…TMPyP complex in the presence of 0 and 30 μM BSA (inset: the complex fluorescence decay curves, λ_ex_ = 455 nm, λ_em_ = 650 nm).

**Figure 4 nanomaterials-11-01674-f004:**
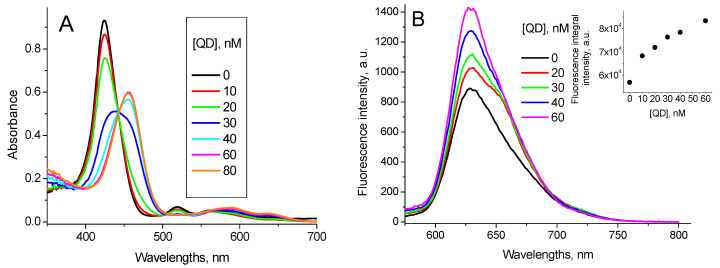
Absorption (**A**) and fluorescence (**B**) spectra (*λ*_ex_ = 445 nm) of 5 μM TMPyP solution in the presence of 30 μM BSA at different CdTe-3MPA QD concentrations. (The fluorescence spectra were recalculated subtracting the respective QD emission from the experimental ones).

**Figure 5 nanomaterials-11-01674-f005:**
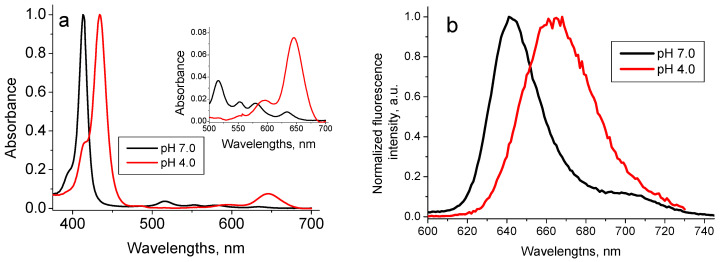
Normalized optical absorption (**a**) and fluorescence spectra (*λ*_ex_ = 420 nm) (**b**) of 1.6 µM TPPS_4_ in the bi-protonated (pH 4.0) and non-protonated (pH 7.0) states [39].

**Figure 6 nanomaterials-11-01674-f006:**
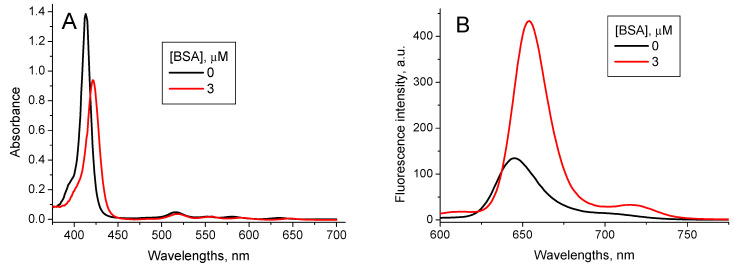
Optical absorption (**A**) and fluorescence (*λ*_ex_ = 417 nm) (**B**) spectra of 1.6 µM TPPS_4_ water solution at pH 7.0 in the absence and presence of 3 µM BSA [37].

**Figure 7 nanomaterials-11-01674-f007:**
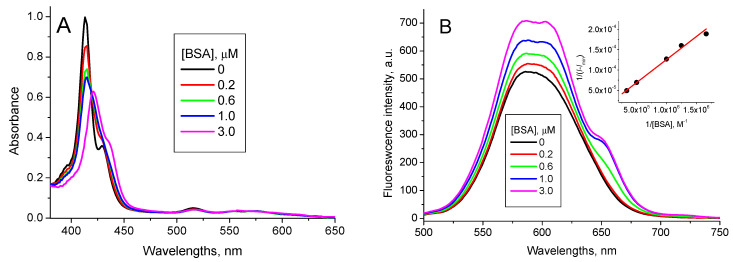
1.6 µM TPPS_4_ optical absorption (**A**) and fluorescence spectra (*λ*_ex_ = 420 nm) (**B**) in the presence of 400 nM CdTe-MPA QD at different BSA concentrations (pH 7.0), (inset: the fitting of the fluorescence integral intensity in accordance with the Equation (7)).

**Figure 8 nanomaterials-11-01674-f008:**
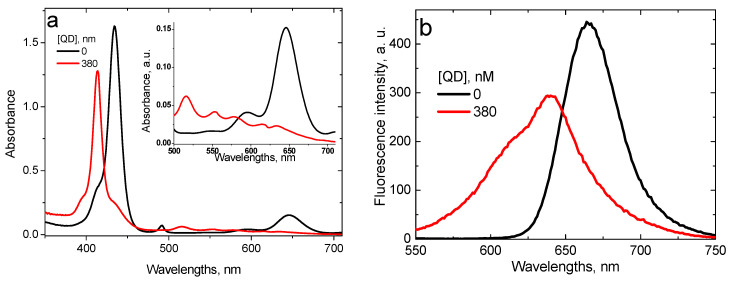
Optical absorption (**a**) and fluorescence (**b**) (λ_ex_ = 420 nm) spectra of the 1.6 μM TPPS_4_ solution at pH 4.0 in the absence and in the presence of 380 nM CdTe-3-MPA QD [29].

**Figure 9 nanomaterials-11-01674-f009:**
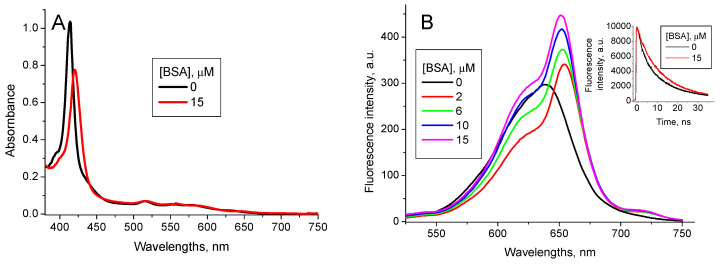
TPPS_4_ optical absorption (**A**) and fluorescence (**B**) spectra (*λ*_ex_ = 417 nm) at pH 4.0 in the presence of 380 nM CdTe-3-MPA QD for different BSA concentrations (Inset: the fluorescence decay curves in the absence and presence of 15 μM BSA, *λ*_ex_ = 420 nm, *λ*_em_ = 650 nm).

**Figure 10 nanomaterials-11-01674-f010:**
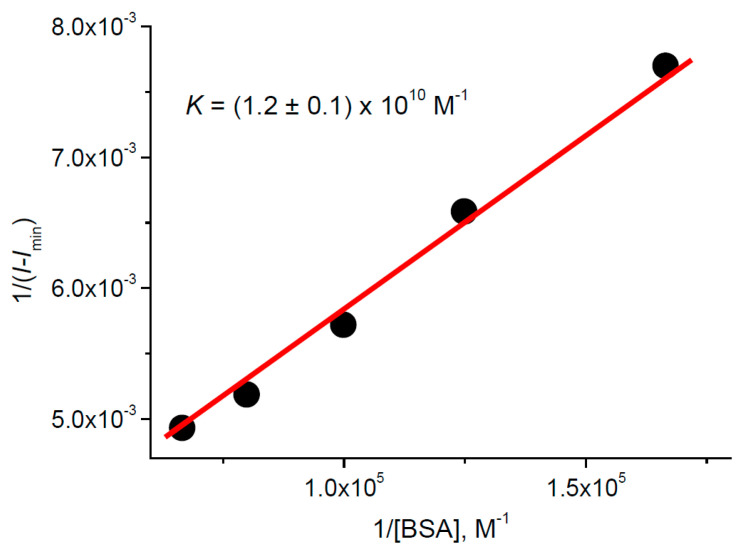
Fitting in accordance with the Equation (3) of the TPPS_4_ fluorescence intensity in the presence of 380 nM CdTe-3-MPAQD in function of BSA concentration at pH 4.0.

**Table 1 nanomaterials-11-01674-t001:** Lifetimes of the short-lived (*τ*_1_) and long-lived (*τ*_2_) components of CdTe-3-MPA QD luminescence decay curves and their relative amplitudes (*I*_1_, *I*_2_) for different BSA concentrations.

[BSA], μM	*τ*_1_, ns	*I* _1_	*τ*_2_, ns	*I* _2_
0	0.913 ± 0.004	0.196 ± 0.002	20.5 ± 0.02	0.804 ± 0.008
2	0.867 ± 0.004	0.169 ± 0.002	21.4 ± 0.02	0.831 ± 0.008
4	0.920 ± 0.003	0.176 ± 0.002	19.6 ± 0.01	0.824 ± 0.008
6	0.903 ± 0.004	0.156 ± 0.002	18.9 ± 0.03	0.844 ± 0.008
8	0.867 ± 0.005	0.160 ± 0.002	18.0 ± 0.03	0.841 ± 0.008
10	0.888 ± 0.005	0.148 ± 0.002	21.5 ± 0.04	0.852 ± 0.008
15	0.922 ± 0.003	0.157 ± 0.002	20.4 ± 0.03	0.843 ± 0.008

## Data Availability

All original data related in this study are presented in the manuscript text and in cited articles.

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
