# Peer review of "Effect of Serum Albumin on Porphyrin-Quantum Dot Complex Formation, Characteristics and Spectroscopic Analysis"

_nanomaterials, 2021, doi:10.3390/nano11071674_

Round 1

Reviewer 1 Report

The paper submitted by Borissevitch and coworkers reports the investigation of the effect of BSA upon the interaction between a specific quantum dot species and two different charged porphyrin derivatives, namely the cationic TMPyP and the anionic TPPS4. The mutual interactions were analyzed at different pH values and studied by spectrophotometric and fluorometric techniques. Although a lot of data have been reported in the text, I found that the authors listed them without a critical approach and deep considerations, especially from a chemical point of view. Indeed in my opinion, a crucial point for data interpretation is the interaction between BSA and QD at pH 4.0 and 7.0 which is completely omitted in the work (something is reported in a paragraph but specific pH values are not indicated). Especially for BSA, the pH value is fundamental to determine the net charge of the species and, considering that its isoelectric point (IP) is around 4.7, the authors have to consider that this species has a positive charge at pH 4 and a negative one at neutral pH (the IP for BSA is never brought into play during the discussion). This feature must to be taken into account during all the considerations in the text and should help the authors to explain the differences observed in the case of the BSA influence on the anionic porphyrin-QD interaction depending from pH. Moreover, a comparison between the data obtained in the case of the two different porphyrin derivatives should be done, since the differences are surely explainable with electrostatic considerations.

On the basis of these considerations, I strongly suggest to completely revised the manuscript, paying also attention to the editing and the material and methods section: a more detailed description of the experimental parameters used in the complexation studies should be introduced, especially to what concerns the molar ratios between the molecular probes used.

Concluding , I cannot consider the manuscript suitable for publication in the present form since major revisions are necessary. Further, I invite the authors to submit the revised form to a more specialized journal, since in my personal opinion these studies don’t properly fit with the Nanomaterials aim and scope.

Author Response

Answers to Reviewer 1 comments

Dear Reviewer!

Thank you for your thorough and competent analysis of our work and useful comments and considerations, which we have studied carefully and introduced the appropriate changes into the text of the article. The responses to your comments are below.

Comment.

In my opinion, a crucial point for data interpretation is the interaction between BSA and QD at pH 4.0 and 7.0 which is completely omitted in the work (something is reported in a paragraph but specific pH values are not indicated). Especially for BSA, the pH value is fundamental to determine the net charge of the species and, considering that its isoelectric point (IP) is around 4.7, the authors have to consider that this species has a positive charge at pH 4 and a negative one at neutral pH (the IP for BSA is never brought into play during the discussion). This feature must to be taken into account during all the considerations in the text and should help the authors to explain the differences observed in the case of the BSA influence on the anionic porphyrin-QD interaction depending from pH.

Answer.

We agree that the electrostatic interaction should play important role in interaction of porphyrins with quantum dots and albumins. This was the principal reason to select one positive and one negative porphyrin for this study.

We agree also that the electrostatic interaction should depend on pH due to changes in net charges of albumin, QD and porphyrin.

Basing on these considerations and in accordance with your recommendations we tried to explain the results, obtained in this study.

However, although electrostatic effects are important in interactions of charged systems, other types of interactions can affect it, as well.

In addition, we would like to note that the detailed and profound study of complex system of porphyrin + QD + albumin should result not in an article but in a monography. In our study we don´t pretend to explain the complete picture of the effect of albumin upon interaction of quantum dots with porphyrins. The interaction between quantum dots and porphyrins itself is already a very complicated problem, various aspects of which have been studied and published in a number of articles. The same can be said about interaction of porphyrins with albumins. Therefore, we treat this study as a first step in investigation of the complex system QD + porphyrin + albumin, the purpose of it is to demonstrate that the effect of albumin on the quantum dot interaction with porphyrins really exists and to show the importance of some factors, such as porphyrin charge and pH on this effect. We believe that the obtained data are sufficient to do some conclusions about albumin effects on the QD interaction with porphyrins.

We would like also to note, that since this work is related to possible application of the studied systems in photochemotherapy, the investigation of their spectroscopic characteristics seems to be especially interesting and important.

Reviewer 2 Report

Iouri Borissevitch and co-workers reported the effect of BSA on CdTe-3-MPA QD-porphyrin complexes. This manuscript is premature as its first part yet to be published (ref 22 submitted for publication). The authors claimed that the effect of SBA on the porphyrin-QD complexation is present based on the observation of the spectral changes. This is not enough to justify their claim. More various scientific evidences are needed to uncover the effect of SBA on the porphyrin-QD complexation in more detail. Therefore, I cannot recommend the publication of this manuscript in Nanomaterials.

Some other comments.

  1. CdTe-3-MPA QD is not a discrete molecule with a specific molecular mass. How did the authors determine the molar concentration of QD species? The concentration should be expressed in weight per volume such as mg/mL. The concentration of SBA should be also expressed in the same way. The values and units for the binding constants will be changed additionally.
  2. Amend the figures (2a, 2c, 3, 4, 5, 6a, 8, 9a, and 10). The numbers on the axes are merged with the axes.
  3. Figure 5, Figure 6 are already published in ref 31 and 29. Either remove or include it in supplementary materials.
  4. Figure 8 are already in ref 22. Delete it.
  5. What does “liquid charge” mean? Why is the term “liquid” necessary?
  6. Correct the reference format (few mistakes).

Author Response

Answers to Reviewer 2 comments

Dear Reviewer!

Thank you for a thorough analysis of our work and useful comments, which we have studied carefully and introduced the appropriate changes to the text of the article. The responses to your comments are below.

Comment 1.

This manuscript is premature as its first part yet to be published (ref 22 submitted for publication).

Answer.

The article refereed as [22] has already been published as [A.L.S. Pavanelli, L.N.C. Máximo, R.S. da Silva, I. Borissevitch, Interaction between TPPS4 porphyrin and CdTe-3-MPA quantum dot: Proton and energy transfer, J. Lumin., 237 (2021) 118213. https://doi.org/10.1016/j.jlumin.2021.118213.]

This information was included in the list of references.

Comment 2.

The authors claimed that the effect of SBA on the porphyrin-QD complexation is present based on the observation of the spectral changes. This is not enough to justify their claim. More various scientific evidences are needed to uncover the effect of SBA on the porphyrin-QD complexation in more detail.

Answer.

We agree that there exist various experimental techniques, which can be used for the detailed study of complex systems, such as in our case quantum dots + porphyrin + albumin. Actually, the detailed study of such complex system should result not in an article but in a monography. In our study we don´t pretend to explain the complete picture of the effect of albumin upon interaction of quantum dots with porphyrins. Just the interaction between quantum dots and porphyrins is already a very complicated problem, which has been studied and published in a number of articles. The same can be said about interaction of porphyrins with albumins. Therefore, we treat this study as a first step in investigation of the complex system QD + porphyrin + albumin, the purpose of it is to demonstrate that the effect of albumin on the quantum dot interaction with porphyrins really exists and to show the importance of some factors, such as the porphyrin charge and pH for this effect. We believe that spectroscopic methods are powerful enough to achieve this purpose and the obtained data are sufficient to do some conclusions about albumin effects on the QD interaction with porphyrins. Anyway, with this in mind we have modified the title of this article not to mislead potential readers.

Comment 3

CdTe-3-MPA QD is not a discrete molecule with a specific molecular mass. How did the authors determine the molar concentration of QD species? The concentration should be expressed in weight per volume such as mg/mL. The concentration of SBA should be also expressed in the same way. The values and units for the binding constants will be changed additionally.

Answer

We agree that QD is not a discrete system with the defined molecular weight. However, this is a common practice to present QD concentration in Mole units, using M = (average molar weight/liter). See, for example, Yan, R., Yu, B. Q., Yin, M. M., Zhou, Z. Q., Xiang, X., Han, X. L., Liu, Y., & Jiang, F. L. (2018). The interactions of CdTe quantum dots with serum albumin and subsequent cytotoxicity: the influence of homologous ligands. Toxicology research7(2), 147–155. https://doi.org/10.1039/c7tx00301c; G. R. Bardajee, Z. Hooshyar, M. Rezaei, M. K. Khaneghah, F. Fallahnejad (2017), Spectroscopic studies on the interactions of capped CdS quantum dots with human serum albumin (HSA) and bovine serum albumin (BSA), Inorganic and Nano-Metal Chemistry, 47:5, 688-696, DOI: 10.1080/15533174.2016.1186098; Shao, L., Dong, C., Sang, F. et al. Studies on Interaction of CdTe Quantum Dots with Bovine Serum Albumin Using Fluorescence Correlation Spectroscopy. J. Fluoresc. 19, 151 (2009). https://doi.org/10.1007/s10895-008-0396-0.Тhis form of representation makes it easy to compare the binding constants for different compounds, which are usually presented in universal units M-1.

The same can be said about the representation of albumin concentrations.

In our study we have used the optical absorption data to determine concentrations of BSA with molar absorption coefficient e = 4.55x104 M-1cm-1 at l = 280 nm, CdTe-3-MPA QD with e = 1.27x105 M-1cm-1 at l = 557 nm, TMPyP with e = 2.26×105 M-1cm-1 at l = 425 nm and TPPS4 with ε =1.3×104M−1cm−1 at l = 515 nm for its non-protonated form (pH 7.0) and ε = 3.26×104M−1cm−1 at l = 644 nm for its biprotonated form (pH 4.0).

The molar absorption coefficient for CdTe-3-MPA QD was determined using the equations (1) and (2).

We have included this information in the manuscript text.

Comment 4

Amend the figures (2a, 2c, 3, 4, 5, 6a, 8, 9a, and 10). The numbers on the axes are merged with the axes.

Answer.

All figures have been corrected.

Comments 5 and 6

Figure 5, Figure 6 are already published in ref 31 and 29. Either remove or include it in supplementary materials.

Figure 8 are already in ref 22. Delete it.

Answer.

Indeed, the figures 5, 6 and 8 have already been published in our previous articles. However, we believe that the presence of these figures in the text of this manuscript facilitates the comparison of the data for QD+poprphyrin+BSA complex systems with the known ones. We have indicated the respective references in the figures´ legends.

Comment 7.

What does “liquid charge” mean? Why is the term “liquid” necessary?

Answer.

The term “liquid charge” is commonly used in various articles, where a molecule possesses several charged groups in its structure and indicates the total charge of a molecule. The meso-tetra methyl pyridyl porphyrin (TMPyP) has four positively charged lateral groups in its structure. Therefore, total or liquid molecule charge is 4+. The meso-tetrakis(p-sulfonato-phenyl) porphyrin (TPPS4) has four negatively charged groups and in its non-protonated state possesses total (liquid) charge 4-, while in its bi-protonated state the liquid charge is 2-. Anyway, this term is not too essential. We have removed it from the text.

Comment 8

Correct the reference format (few mistakes).

Answer.

We have tried to correct all references.

Reviewer 3 Report

The paper by Pavanelli et al. shows the study of binding two porphyrines (one positively, one negatively to charges) to QDs. The authors also include bovine serum albumin (BSA), binding to porphyrines and QDs, and therefore possibly modifying the QD-porphyrines interaction.

The results may be important and publishable, however, I would suggest several improvements.

First, the text is rather messy and needs reorganization and proofreading by a native speaker. See eg. lines 41-44. There is a lot of other examples. Also, some figures lack ticks and their X-axis, and it makes analysis harder.

The authors are mentioning biologically active window and the possible use of porphyrine-nanoparticle complexes in photodynamic therapy. How this research is connected to this part of introduction? Is there a possibility to use toxic CdTe nanoparticles in photodynamic treatment? Is there any indication, that QD particles are upconvertors? Why then authors did not use QDs absorbing in this biologically active range? Bigger QDs have higher area, and it can influence somehow bidning constants. However, usually different, less toxic nanoparticles are used in photodynamic therapy, and therefore I don’t see a point in giving this example as a potential application.

Equations should be moved to methods and well explained, e.g. equation (5) - what are I1, I2? When you are giving tau values, you should also provide those amplitude values, or at least calculate average tau.

Describing absorption and emission spectra (chapter 3), please provide values of maxima, not only range.

How do you explain broadening in QD emission (main maximum about 560 nm and additional at 600 nm)? It suggests the non-homogenous preparation of QDs.

You don’t have a final proof for claiming at lines 138-140. Change in relative ratio may be the same, if one component in increased or second decreased. Please, also discuss the particular tau values. Were they constant or changed upon addition of BSA?

The Kd value at line 148 was determined at what pH? The authors are using pH 7 and pH 4. What is Kd for the second pH?

Explain the spectral shift showed in Fig.4A. This is due to aggregation/disaggregation? Change in environment polarity?

Please, correct  Fig. 4B emission spectra for QD emission (show QD titration without TMPyP). Now it looks like the increase may be only due to QD addition. Make the similar type of corrections for other spectra shown within the manuscript. Discuss also the contribution of QD emission in porphyrine lifetime measurement. There is a clear spectral overlap, so the controls need to be shown/included.

About the conclusion in lines 194-196 - there is a lot of methods to verify complex stability, not only spectrum measurements. One of the simples is running electrophoresis in agarose gel (see e.g. Fig S18 in the paper by Sławski et al (2021) https://pubs.acs.org/doi/abs/10.1021/acs.jpcb.1c00325. It shows also the QD-BSA complex, and the same method may be used here.

Author Response

Answers to Reviewer 3 comments

Dear Reviewer!

Thank you for the professional and profound analysis of our work and important and useful comments, which we have studied carefully and introduced the appropriate changes to the text of the article. The responses to your comments are below.

Comment 1.

First, the text is rather messy and needs reorganization and proofreading by a native speaker. See eg. lines 41-44. There is a lot of other examples. Also, some figures lack ticks and their X-axis, and it makes analysis harder.

Answer.

We have checked the spelling in the manuscript text and thereafter it has been corrected by a specialist in the area with native English. The figures have been corrected.

Comment 2.

The authors are mentioning biologically active window and the possible use of porphyrine-nanoparticle complexes in photodynamic therapy. How this research is connected to this part of introduction? Is there a possibility to use toxic CdTe nanoparticles in photodynamic treatment? Is there any indication, that QD particles are upconvertors? Why then authors did not use QDs absorbing in this biologically active range? Bigger QDs have higher area, and it can influence somehow bidning constants. However, usually different, less toxic nanoparticles are used in photodynamic therapy, and therefore I don’t see a point in giving this example as a potential application.

Answer.

Due to their high optical absorption in broad spectral region, intensive luminescence and high dark and photostability Quantum dots (QD) attract a special interest for applications in biology and medicine, where they can compete successfully with organic compounds as fluorescent probes and/or photosensitizers (PS) for photochemotherapy. Various studies in this direction have already been realized (see, for example,

M.F. Frasco, N. Chaniotakis, Bioconjugated quantum dots as fluorescent probes for bioanalytical applications. Anal Bioanal Chem 396 (2010), 229–240. doi:10.1007/s00216-009-3033-0;

H.M.E. Azzazy, M.M.H. Mansour, S.C. Kazmierczak From diagnostics to therapy: prospects of quantum dots. Clin Biochem 40 (2007), 917–927. doi:10.1016/j.clinbiochem.2007.05.018;

A.C.S. Samia, X. Chen, C. Burda, Semiconductor Quantum Dots for Photodynamic Therapy, JACS, 125(51) (2003), 15736-15737. DOI: 10.1021/ja0386905;

  1. Yaghini, A.M. Seifalian, A.J. MacRobert, Quantum dots and their potential biomedical applications in photosensitization for photodynamic therapy, NANOMEDICINE (REVIEW), 4(3) (2009) https://doi.org/10.2217/nnm.09.9])

On the other hand, being inorganic semiconductors, QDs manifest high toxicity toward biological objects and possess low water solubility, which restricts their direct application in medicine and biology. However, when the QD surface is covered with appropriate molecules (functionalized QD), it overcomes this imperfection, as in this case living systems appear protected from the contact with the toxic QD core (CdTe or CdSe, for example). Moreover, functionalization increases the QD water solubility and improves their affinity with biological structures:

  1. Zhang, Surface functionalization of quantum dots for biotechnological applications. Adv Colloid Interface Sci, 215 (2011), 28–45. doi:10.1016/j.cis.2014.11.004.

A.S. Karakoti, R. Shukla, R. Shanker, S. Singh, Surface functionalization of quantum dots for biological applications. Adv Colloid Interf Sci 215 (2015), 28–45. https://doi.org/10.1016/j.cis.2014.11.004.

At the same time, the mechanisms of QD action in photochemotherapy, not well studied yet, differ from the photodynamic effect, the basis of the photodynamic therapy (PDT), which is effective and well described. Therefore, it seems interesting to use the beneficial properties of QDs to increase the efficiency of classical photosensitizers.

Among the photosensitizers, porphyrins are the most widely applied in PDT. Nevertheless, a great disadvantage of porphyrins for applications in PDT lies in their relatively low optical absorption in the spectral region of the phototherapeutic window (600-800 nm), where biological tissues are most transparent. The problem could be corrected by porphyrin complexation with certain substances, able to absorb light energy in this spectral range and transfer this energy to porphyrin molecules. Due to their intense absorption in a broad spectral region quantum dots may serve as effective antennas for light energy accumulation, and their intense narrow luminescence band facilitates the energy transfer to corresponding photosensitizer, porphyrin, in particular, thus increasing the efficiency of the light energy utilization and consequently increasing the photosensitizer efficacy. See, for example:

A.C.S. Samia, S. Dayal, C. Burda, Quantum Dot-based Energy Transfer: Perspectives and Potential for Applications in Photodynamic Therapy, Photochem&Photobiol, 82(3) (2006), 617-625. https://doi.org/10.1562/2005-05-11-IR-525;

I.V Martynenko, V.A. Kuznetsova, А.O. Orlova, P.A. Kanaev, V.G. Maslov, A. Loudon, V. Zaharov, P. Parfenov, Yu.K. Gun'ko, A.V. Baranov, Chlorin e6–ZnSe/ZnS quantum dots based system as reagent for photodynamic therapy, Nanotechnology, 26(5) (2015) 055102;

  1. Dayal, R. Królicki, Y. Lou et al, Femtosecond time-resolved energy transfer from CdSe nanoparticles to phthalocyanines. Appl Phys B Lasers Opt 84 (2006), 309–315. https://doi.org/10.1007/s00340-006-2293-z;
  2. Gromova, A.O. Orlova, V.G. Maslov et al, Fluorescence energy transfer in quantum dot/azo dye complexes in polymer track membranes. Nanoscale Res Lett 8 (2013), 452. https://doi.org/10.1186/1556-276X-8-452;

I.E. Borissevitch, E.P. Lukashev, I.P. Oleinikov, A.L.S. Pavanelli, P.J. Gonçalves, P.P. Knox, Electrostatic interactions and covalent binding effects on the energy transfer between quantum dots and reaction centers of purple bacteria, J. Lumin., 207 (2019), 129-136. https://doi.org/10.1016/j.jlumin.2018.11.013].

Thus, we can assert that the study of porphyrin interaction with QD is justified and important for the development of photochemotherapy.On the other hand, being introduced into an organism, porphyrins and QDs should interact with various organized structures, such as cell membranes, nucleic acids, proteins, etc. Among them serum albumin attracts a special attention due to its ability to bind various compounds and transport them within the organism with blood flux. Besides, albumins possess elevated affinity to malignant tissues. This also makes albumins perspective compounds as a delivery system. On the other hand, interaction with albumins can affect complexation of porphyrins with QD, changing their own and complex characteristics. Therefore, we believe that the study of the effects of albumins on the interaction of porphyrins with QDs is of importance for application in PDT and other types of phototherapy.

We hope the above arguments be sufficient to substantiate our study.

Comment 3.

Equations should be moved to methods and well explained, e.g. equation (5) - what are I1, I2? When you are giving tau values, you should also provide those amplitude values, or at least calculate average tau.

Answer.

The equations (4)-(6) are associated with the treatment of data obtained during the experiments and their use cannot be foreseen before the realization of experiments. Therefore, we believe more reasonable to include these equations in the “result” section as they are necessary for discussion and end conclusions.

In the absence of albumin, the QD luminescence decay curve has been successfully fitted as bi-exponential:

                       (4)

with the component lifetimes  = 0.9 ± 0.1 and  = 20 ± 1 ns.

The addition of albumin does not change the bi-exponential character of the decay curves. Moreover, the component lifetimes are independent on the albumin concentration, while their amplitudes vary. The decay curves were obtained via time-correlated single photon counting technique and their total amplitudes were standard (10.000 counts) and independent of real luminescence intensity. Therefore, it seems more correct to characterize the relative contribution of a component not by its absolute amplitude Ii but by relative one Ri, which can be calculated as:

Anyway, if  finally we have .

Indeed, R1 can increase not only with I1 increase, but also with I2 decrease. However, the similarity of R1 dependence on BSA concentration with that of the integral luminescence intensity (Ilum) on [BSA] means that the increase of Ilum is associated with the increase of R1.

In our opinion, the characterization of the luminescence kinetics by an average lifetime is not informative as it does not demonstrate the real character of the dependence of the decay curve on BSA concentration.

Comment 4

Describing absorption and emission spectra (chapter 3), please provide values of maxima, not only range.

Answer.

The corrections were included in the text.

Comment 5

How do you explain broadening in QD emission (main maximum about 560 nm and additional at 600 nm)? It suggests the non-homogenous preparation of QDs.

Answer.

In our recent study [22, A.L.S. Pavanelli, L.N.C. Máximo, R.S. da Silva, I. Borissevitch, Interaction between TPPS4 porphyrin and CdTe-3-MPA quantum dot: Proton and energy transfer, J. Lumin., 237 (2021) 118213. https://doi.org/10.1016/j.jlumin.2021.118213] we have demonstrated that the broadening in QD emission is due to existence of two QD forms: with protonated and with non-protonated functionalization groups. The first one has the luminescence maximum at 620 nm and the second one - at 578 nm. Thus, this broadening is not the result of the non-homogenous QD preparation.

Comment 6.

You don’t have a final proof for claiming at lines 138-140. Change in relative ratio may be the same, if one component in increased or second decreased. Please, also discuss the particular tau values. Were they constant or changed upon addition of BSA?

Answer.

See answer to Comment 3. Besides, we can say the follow. Recently it has been shown [13, 19-22], that the short-lived CdTe-3-MPA QD luminescence component is associated with the electron-hole annihilation in the QD core, while the long-lived one is due to the electron-hole annihilation in the QD functionalizing (protective) shell. Therefore, the increase of the QD luminescence intensity in the presence of BSA can be due to interaction of 3-MPA groups in the QD surface with BSA. This can be associated with the increase in rigidity of the QD surface groups at their binding with BSA.

We have tried to give more detailed consideration in the text.

Comment 7.

The Kd value at line 148 was determined at what pH? The authors are using pH 7 and pH 4. What is Kd for the second pH?

Answer.

Excuse our incorrectness.

The presented results were obtained at pH 7.0. At pH 4.0 we have not observed any significate effect of BSA on QD characteristics at the used concentration range. We explain this fact by strong reduction of the probability of QD binding with BSA due to their strong repulsion at pH 4.0. Indeed, the isoelectric point of BSA is at pH 4.7. At the same time the pKa point of CdTe-3-MPA QD functional groups is at pH 4.34 [Industrial hygiene and toxicology. Vol. II. 2nd rev. ed. Frank A. Patty, Editor. John Wiley & Sons, Inc., 605 Third Ave., New York 16, N. Y., 1963. https://doi.org/10.1002/jps.2600520932]. Thus, at pH 4.0 the total charge of both BSA and QD, is positive, which should reduce the probability of their binding making the binding constant less than (1.2 ±0.1) x 106 M-1.

Comment 8.

Explain the spectral shift showed in Fig.4A. This is due to aggregation/disaggregation? Change in environment polarity?

Answer.

The appearance of the new peak at 455 nm is associated with a charge transfer complex between TMPyP porphyrin and QD. Thus, we can affirm that BSA in the solution does not prevent the formation of the CdTe-3MPA…TMPyP complex. As it has been demonstrated above, the addition of BSA to the CdTe-3MPA…TMPyP solution, with the complex already formed, dos not destruct this complex.

This could be explained by the fact that the constant of the CdTe-3MPA…TMPyP complex formation is approximately 10 times larger than that for TMPyP binding with BSA and 5 times larger than for CdTe-3MPA binding with BSA. Therefore, in the system of equilibria

CdTe-3MPA + TMPyP Û CdTe-3MPA…TMPyP       KTMPyP_QD = (6.0 ± 0.5) x 106 M-1

BSA + TMPyP Û BSA…TMPyP                KTMPyP_BSA = 7.3 x 105 M-1

CdTe-3MPA + BSA Û CdTe-3MPA…BSA          KQD_BSA = (1.2 ± 0.1) x 106 M-1

the first one should be predominant.

Comment 9.

Please, correct  Fig. 4B emission spectra for QD emission (show QD titration without TMPyP). Now it looks like the increase may be only due to QD addition. Make the similar type of corrections for other spectra shown within the manuscript. Discuss also the contribution of QD emission in porphyrine lifetime measurement. There is a clear spectral overlap, so the controls need to be shown/included.

Answer.

The presented fluorescence spectra have been recalculated subtracting the respective QD emission from the experimental ones. We have added the explanation to the text of the figure legend. The QD emission spectra have been shown on the Figure 2B.

Comment 10.

About the conclusion in lines 194-196 - there is a lot of methods to verify complex stability, not only spectrum measurements. One of the simples is running electrophoresis in agarose gel (see e.g. Fig S18 in the paper by Sławski et al (2021) https://pubs.acs.org/doi/abs/10.1021/acs.jpcb.1c00325. It shows also the QD-BSA complex, and the same method may be used here.

Answer.

Thank you for the indicated reference. We have included it into the list of references.

We agree that there exist various experimental techniques, which can be used for the detailed study of complex systems, such as in our case quantum dots + porphyrin + albumin. Actually, the detailed study of such complex system should result not in an article but in a monography. In our study we don´t pretend to explain the complete picture of the effect of albumin upon interaction of quantum dots with porphyrins. Just the interaction between quantum dots and porphyrins is already a very complicated problem, which has been studied and published in a number of articles. The same can be said about interaction of porphyrins with albumins. Therefore, we treat this study as a first step in investigation of the complex system QD + porphyrin + albumin, the purpose of it is to demonstrate that the effect of albumin on the quantum dot interaction with porphyrins really exists and to show the importance of some factors, such as porphyrin charge and pH on this effect. We believe that spectroscopic methods are powerful enough to achieve this purpose and the obtained data are sufficient to do some conclusions about albumin effects on the QD interaction with porphyrins. Anyway, with this in mind we have modified the title of this article not to mislead potential readers.

Round 2

Reviewer 1 Report

I'm satisfied with the changes made by the authors. Just two further observations: 

  • Figure 1: improve the image quality, it is of very low resolution
  • Figure 7: it's missing in the submitted pdf

Author Response

Answers to Reviewer 1 comments

Dear Reviewer!

Thank you for your comments.

Comment 1.

Figure 1: improve the image quality, it is of very low resolution

Answer.

Figure 1 was remade.

Comment 2.

Figure 7: it's missing in the submitted pdf

Answer.

Figure 7 was remade and now it is present in the text both in docx and in pdf.

Reviewer 2 Report

The manuscript has been much improved and is acceptable after minor revision.

1) line 82: Change "synthetized" into "synthesized".

2) Figure 7 is missing.

Author Response

Answers to Reviewer 2 comments

Dear Reviewer!

Thank you for your comments.

Comment 1.

line 82: Change "synthetized" into "synthesized"

Answer.

The correction was made.

Comment 2.

Figure 7 is missing.

Answer.

Figure 7 was remade.

Reviewer 3 Report

I'm satisfied with the changes, introduced by the authors. 

Just one point for reconsideration - I understand the reasons for analysing the relative ratio of amplitudes, not average tau. However, maybe you could also provide the amplitude values, to allow comparison with other studies? Mathematically, it could be deduced from the ratio, but a direct indication might be better for some readers.

Fig. 7 is missing in my copy of the manuscript. Please, check.

Author Response

Answers to Reviewer 3 comments

Dear Reviewer!

Thank you for your comments.

Comment 1.

Just one point for reconsideration - I understand the reasons for analysing the relative ratio of amplitudes, not average tau. However, maybe you could also provide the amplitude values, to allow comparison with other studies? Mathematically, it could be deduced from the ratio, but a direct indication might be better for some readers.

Answer.

We have included Table 1in the text, which shows the lifetimes and amplitudes of the QD luminescence decay curves for different BSA concentrations. These values were obtained due to the bi-exponential fitting of respective curves.

Comment 2.

Fig. 7 is missing in my copy of the manuscript. Please, check.

Answer.

Figure 7 was remade and now is present in the text both in docx and in pdf.